# The Cytotoxicity of Carbon Nanotubes and Hydroxyapatite, and Graphene and Hydroxyapatite Nanocomposites against Breast Cancer Cells

**DOI:** 10.3390/nano13030556

**Published:** 2023-01-30

**Authors:** Tristan Nguyen, Anuj Maniyar, Mrinmoy Sarkar, Tapasree Roy Sarkar, Gururaj M. Neelgund

**Affiliations:** 1Department of Biology, Texas A&M University, College Station, TX 77843, USA; 2Department of Chemistry, Prairie View A&M University, Prairie View, TX 77446, USA

**Keywords:** hydroxyapatite, carbon nanotubes, graphene, cytotoxicity, breast cancer

## Abstract

Cancer is a current dreadful disease and the leading cause of death. Next to cardiovascular diseases, cancer is the most severe threat to human life and health. Breast cancer is the most common invasive cancer diagnosed in women. Each year about 2.3 million women are diagnosed with breast cancer. In consideration of the severity of breast cancer, herein we designed the biocompatible nanomaterials, CNTs-HAP and GR-HAP, through grafting of hydroxyapatite (HAP) to carbon nanotubes (CNTs) and graphene (GR) nanosheets. CNTs-HAP and GR-HAP have been tested for their cytotoxicity, growth and motility inhibitory effects, and their effects on the mesenchymal markers. All these demonstrated significant dose-dependent and time-dependent in vitro cytotoxicity against SUM-159 and MCF-7 breast cancer cell lines. The cell viability assay showed that the CNTs-HAP was more effective over SUM-159 cells than MCF-7 cells. It found that the increase in the concentration of GR-HAP has inhibited the clonogenic ability of breast cancer cells. The GR-HAP exhibited a substantial inhibitory effect on the cell motility of SUM-159 cell lines. It was investigated that the expression of vimentin (mesenchymal marker) was majorly reduced in SUM-159 cells by GR-HAP.

## 1. Introduction

Cancer is known to be a leading cause of death worldwide [1]. It has been recognized by the World Health Organization (WHO) that cancer is a dreadful disease and is the most common cause of death [2]. Next to cardiovascular diseases, cancer is the most severe threat to human life and health [3]. Although significant progress has been achieved with cancer research and treatment, the incidence of cancer is still rising. Among different categorized cancers, breast cancer is the deadliest for women and contributes to the highest mortality [4,5,6]. Breast cancer is the most common invasive cancer diagnosed in women [2]. Despite enhanced preventive measures and treatment strategies, the risk of relapse remains as high as 20–30%, with 5–7% of breast cancer patients developing the metastatic disease later on in their lives [7,8]. Each year about 2.3 million women are affected by breast cancer, and it is estimated that approximately 685,000 women died in 2020 from this cancer [9,10]. The possible origin of breast cancer is estrogen receptor beta (ERβ) [4]. About 60% of breast cancer cases are related to hormones and are estrogen-dependent [4,11,12,13]. The shortfall of estrogen receptor alpha/progesterone receptor expression can lead to a subtype of breast cancer called triple-negative breast cancer (TNBC) [4,12,14]. About 90% of deaths from breast cancer are caused by metastasis [5,6]. Although progress has been made in cancer treatment and techniques, breast cancer still has high mortality among women [1,15].

Therefore, the development of new therapeutic strategies is urgently needed for the efficient treatment of breast cancer. Nanotechnology is an ideal platform for developing emerging cancer therapies [16]. The nanomaterial-instigated therapies have distinct advantages and are best promises to target cancer cells more precisely with reduced side effects [16,17]. The nanomaterials exhibit unique properties and interesting structures due to their small size, which facilitates the targeting of cancer cells and their facile admittance into cancer cells [16]. In addition, it provides the freedom of conjugating nanomaterials with secondary materials to acquire desired properties [16,17,18]. Further, the surface of nanomaterials could be modified to the requirement using specified ligands and receptors [16,17,18]. Overall, nanomaterials have the ability to cross biological barriers with minimal side effects [19,20,21]. 

Hydroxyapatite (HAP) is a vital biomaterial having a chemical composition, mineral constituent, and crystallographic structure identical to the inorganic component of hard tissues and dentine [22,23,24]. HAP offers great opportunities for cancer therapy due to its distinctive intrinsic antitumor ability and excellent biocompatibility [25,26,27,28]. HAP plays a significant role in cancer therapy as the cancer cells are spread over the skeletal tissues containing HAP [29]. HAP is capable of inhibiting the growth and metastasis of tumor cells [30]. The significant cytotoxicity of HAP against breast, osteosarcoma, and gastric cancer cells has been revealed [31,32]. The induction of apoptosis could cause the cytotoxicity of HAP in cancer cells through the mitochondria-dependent pathway that ensues from oxidative stress or inhibition of protein synthesis due to abundant internalized HAP in cancer cells around the endoplasmic reticulum [31,33]. HAP-induced in vitro MTT assay demonstrated lower proliferation of A875 cells [27]. It could prevent the local recurrence of malignant melanoma also [27]. Overall, HAP provides excellent cancer therapy opportunities and could be a smart anti-cancer agent [25,26]. However, the efficiency of HAP could be further improved by its surface modification with secondary materials to facilitate the targeting and killing of cancer cells without harming normal or healthy cells. For which carbon allotropes viz., carbon nanotubes (CNTs), and graphene (GR) are idealistic nanomaterials.

CNTs possess a distinctive one-dimensional structure originated by rolling a thick sheet of graphene nanosheets into a smooth cylinder with a high aspect ratio [34,35,36,37,38,39]. CNTs have lightweight with high mechanical strength, unique electronic properties, large surface area, high chemical stability, and excellent thermal stability [40]. The needle-like shape of the CNTs would facilitate their entry into the targeted cancer cells through a different mechanism of passive diffusion across the lipid bilayer or endocytosis, whereby the CNTs bind to the surface of the cancer cell and are, subsequently, engulfed by the cell [41,42]. Moreover, the sharp-edged architecture of CNTs promotes tumor penetration and is ideal for the targeted delivery of anti-cancer drugs [43]. Additionally, CNTs can carry therapeutic agents to many intracellular targets, such as the nucleus, mitochondria, and cytoplasm, or target the TME components to disturb tumor cells’ living conditions, both of which can result in an enhanced antitumor effect [43]. The high specific surface area of CNTs enables to conjugation of various therapeutic molecules [44].

GR is another interesting and the most promising nanomaterial of the 21st century [45]. It has a distinct two-dimensional structure consisting of a single layer of carbon atoms arranged in a honeycomb fashion [45]. Its 2D plane containing sp2 hybridized carbon atoms resulted in delocalized out-of-plane π-bonds afford exceptional carrier mobility [46]. GR is associated with a unique combination of physiochemical properties, such as high surface area, optimal thermal conductivity, remarkable optical transparency, strong mechanical strength, and room temperature quantum hall effect [47,48,49,50,51]. It has an intrinsic photoluminescent property crucial for imaging and tracking cancer therapy [45]. GR has high photothermal properties and oxidative reactivity for drug delivery [19]. The layer structure of GR permits a high loading capacity for therapeutic agents [45]. Therefore, the physical and chemical properties of CNTs and GR are perfectly suitable and particularly important in precisely targeting cancer cells [52]. The CNTs and GR could be rapidly cleared from the body [45]. Furthermore, the strong absorption of CNTs and GR in the near-infrared (NIR) region and their high photothermal conversion efficiency are well suited for photothermal therapy (PTT) of cancer. 

Hence, considering their essential properties and ideal suitability for cancer therapy, we functionalized the CNTs and GR by grafting with HAP. The produced HAP grafted CNTs (CNTs-HAP) and GR-conjugated HAP (GR-HAP) have exhibited interesting properties required for cancer therapy. Although previous studies showed the anti-cancer properties of HAP nanoparticles and CNT, the role of GR -HAP and CNT-HAP is not yet known. Our work demonstrated considerable anti-cancer capabilities of the examined CNTs-HAP and GR-HAP using breast cancer cell lines. 

## 2. Materials and Methods

### 2.1. Experimental Section

All the reagents were used without any purification unless otherwise noted, and the aqueous solutions were prepared using ultrapure water obtained from the Milli-Q Plus system (Millipore; Burlington, MA, USA). CNTs prepared by chemical vapor deposition (CVD) were obtained from Carbon Nanotechnology Laboratory at Rice University, Houston, TX, USA. The graphene oxide was prepared from graphite powder (<20 μm, Sigma Aldrich; St. Louis, MO, USA).

### 2.2. Preparation of CNTs-HAP 

CNTs-HAP was prepared according to the reported procedure [36,53]. In brief, 100 mg carboxylated CNTs (CNTs-COOH), obtained by refluxing the CNTs in a mixture of 1:3 (*v/v*) nitric acid and sulfuric acid, was dispersed in 25 mL DI water. Then, 0.01 mol/L aqueous solution of calcium hydroxide was added, and the resulting mixture was allowed to stir for an hour at ambient conditions. The pH of the mixture was adjusted to 9 using phosphoric acid, and the mixture was stirred at room temperature for 30 min. Thus, formed CNTs-HAP was collected by centrifugation, purified by washing with DI water, and dried under a vacuum.

### 2.3. Preparation of Graphene Oxide

Graphene oxide was prepared from graphite powder according to the Hummers and Offeman method with slight modifications [51,54]. In a typical procedure, 1 g of graphite powder was added to 40 mL concentrated H_2_SO_4_ (Sigma Aldrich; St. Louis, MO, USA) and stirred for 1 h under ice-cooling conditions. Then, 15 mL fuming HNO_3_ (Sigma Aldrich; St. Louis, MO, USA) was added slowly, stirring the mixture for 30 min. To this, 5 g of KMnO_4_ (Sigma Aldrich; St. Louis, MO, USA) was gradually added with stirring and cooling. The resulting mixture was stirred at room temperature for 12 h, followed by the addition of 150 mL DI water. After stirring the mixture for 30 min, 30 mL of H_2_O_2_ (30%) (Sigma Aldrich; St. Louis, MO, USA) was slowly added, which turned the color of the reaction mixture to bright yellow. The resulting mixture was centrifuged and washed with 1:10 HCl (Sigma Aldrich; St. Louis, MO, USA) in the water solution to remove metal ions. Further, the mixture was washed with DI water until complete removal of the acids, and thus obtained dark-yellow colored GO was dried under vacuum at 40 °C for 12 h. The drying process of GO was carried out at a lower temperature to avoid deoxygenation.

### 2.4. Preparation of GR

Graphene oxide (50 mg) was dispersed in EtOH (50 mL) by sonication for 5 mins, the dispersion was subjected to centrifugation, and EtOH was removed [55]. The obtained graphene oxide was redispersed in ethylenediamine (EDA) (Sigma Aldrich; St. Louis, MO, USA) to form the yellow–brown suspension, which was subjected to refluxing for 1 h at 80 °C to yield N-doped GR [56,57]. The resulting GR was centrifuged, washed with EtOH and DI water, and dried under vacuum at 40 °C for 5 h.

### 2.5. Preparation of GR-HAP Nanocomposite 

GR-HAP was synthesized using the reported procedure [53,55]. In detail, 40 mg GR was dispersed in 40 mL DI water through sonication for 5 min. An aqueous solution of 0.01 mol L^−1^ of calcium hydroxide (Sigma Aldrich; St. Louis, MO, USA) was slowly added. The resulting suspension was stirred for an hour in ambient conditions. Then, pH of the mixture was adjusted to 9 with phosphoric acid (Sigma Aldrich; St. Louis, MO, USA) and allowed to stir for 30 min. The resulting GR-HAP was separated by centrifugation, washed with DI water, and dried under a vacuum. The preparation of CNTs-HAP and GR-HAP has been schematically depicted in Appendix A.

### 2.6. Cell Culture

Human breast cancer cell lines, viz., MCF-7 (ATCC HTB-22) and SUM-159, obtained as a gift from MD Anderson Cancer Center, were cultured using the reported procedure [58]. Precisely, MCF-7 cells (ER^+^ cell line) and SUM-159 (ER^™^/ PR^™^/ HER2^™^, or TNBC) cell lines were grown in Dulbecco's Modified Eagle’s Medium (DMEM) (ATCC; Manassas, VA, USA) supplemented with 10% Fetal Bovine Serum (FBS) (ATCC; Manassas, VA, USA), 5 mM glutamine, 0.5% penicillin–streptomycin (Gibco™; Thermo Fisher Scientific, Inc., Waltham, MA, USA), and MCF-7 was grown with additional 5 mM sodium pyruvate (Gibco™; Thermo Fisher Scientific, Inc., Waltham, MA, USA). They were then incubated (PHCbi; Wood dale, IL, USA), maintaining a 5% CO_2_ atmosphere at 37 °C. The cells were cultured by regular splitting and avoiding over-confluency. The cells were washed with PBS (Gibco™; Thermo Fisher Scientific, Inc., Waltham, MA, USA) with pH 7.4 and detached from the flasks using 0.05% trypsin-EDTA (Gibco™; Thermo Fisher Scientific, Inc., Waltham, MA, USA) during subculture and harvesting.

### 2.7. MTT Assay

The cytotoxicity of CNTs-HAP and GR-HAP was analyzed by MTT cell proliferation assay kit (Cayman Chemical Company, #10009365; Ann Arbor, MI, USA) with SUM-159 and MCF-7 cells using the manufacturer’s standard protocol [59]. The CNTs-HAP and GR-HAP were dispersed in a cell culture medium to obtain a 200 μg/mL (ppm) solution, and the serial dilutions were made to get 5, 25, and 50 μg/mL solutions in cell culture media. The cells were seeded at a density of 8 × 10^3^ cells/well of 96-well plates (Thermo Fisher Scientific; Waltham, MA, USA) and were incubated overnight in a 5% CO_2_ atmosphere at 37 °C to allow attachment to the plate. The media was replaced the following day using CNTs-HAP and GR-HAP at different concentrations along with the control sample. Following 24 h of incubation with treatments, 10 μL of MTT reagent was added to each well and subjected to incubation for 4 h. After confirming the formation of the purple-colored formazan crystals under a microscope, the crystals were dissolved with 100 μL of crystal-dissolving solution (Cayman Chemical Company, #10009365; Ann Arbor, MI, USA). The plate was incubated overnight in the dark at room temperature before recording the absorbance at 570 nm using a spectrophotometric plate reader (Molecular Devices; San Jose, CA, USA).

### 2.8. Colony Formation Assay 

MCF7 and SUM-159 cells were seeded (200 cells/well) in 6-well plates (Thermo Fisher Scientific; Waltham, MA, USA) and incubated overnight at 37 °C [60]. The following day, cells were incubated with CNTs-HAP and GR-HAP at the required concentration. The culture medium containing nanoparticles was substituted with fresh medium after 48 h of treatment. The medium was changed every week for the next two weeks. After two weeks, cells were washed twice with PBS, fixed with cold methanol (EMD Millipore; Burlington, MA, USA) (60% in distilled water (*v/v*)) for 30 min at 4 °C, and stained with crystal violet dye (Aldon Corporation; Avon, NY, USA) (0.1% *w/v*) at room temperature for 1 h. Then, the plates were washed with water, dried, and the colonies were counted.

### 2.9. Scratch Assay

The scratch assay was performed by seeding the cells (~3 × 10^5^ cells/well) in 6-well plates and allowing 8–10 h to adhere in a medium containing 0.5% fetal bovine serum (FBS) [58]. A scratch was made using a sterile pipette tip which filled the dish with fresh medium. Then, the CNTs-HAP and GR-HAP were added at the required concentration. Moreover, the wound closure experiment was monitored and recorded for the next 24 h.

### 2.10. Immunofluorescence Assay

The immunofluorescence assay was carried out as described previously [61]. The SUM-159 cells (3.5 × 10^5^ cells/mL) were seeded on a coverslip in a 6-well plate and incubated for 24 h at 37 °C with 5% CO_2_. Then, the cells were treated with the CNTs-HAP and GR-HAP (control group) for 24 min and immediately fixed with 4% paraformaldehyde (Santacruz Biotechnology Inc.; Dallas, TX, USA) at room temperature for 20 min. The primary antibody used for the immunofluorescence assay was Vimentin (D21H3) XP^®^ Rabbit mAb (Cell signaling Technology, #5741S, 1:1000; Danvers, MA, USA), and the secondary antibody was anti-rabbit IgG (Cell Signaling Technology, #4412S, 1:100; Danvers, MA, USA). The nuclei were visualized by DAPI (Prolong Gold Antifade with DAPI, Molecular Probes, #8961S; Thermo Fisher Scientific, Waltham, MA USA), and the images were captured by Echo Rebel microscope (Echo, San Diego, CA, USA).

### 2.11. Characterization

ATR-IR spectra of samples were recorded using a Smiths ChemID diamond attenuated total reflection (DATR) spectrometer (Smiths Detection, Inc., London, United Kingdom), and the powder XRD patterns were collected by Scintag X-ray diffractometer (PAD X) (Cupertino, CA, USA) equipped with Cu Kα photon source (45 kV, 40 mA) at a scanning rate of 3°/min. The SEM images were captured with a JEOL JXA-8900 microscope (JEOL USA, Inc., Peabody, MA, USA).

### 2.12. Statistical Analysis

All the data regarding the treatment of the cells have been obtained from experiments done in triplicate. The statistical analysis was done by one-way, two-way ANOVA, or paired *t*-tests, wherever required, using GraphPad Prism (version 8.0). Data were represented as mean ± S.D. (standard deviation). A *p*-value less than at least 0.05 (* *p* < 0.05) was considered significant.

## 3. Results

### 3.1. Preparation of the CNTs-HAP and GR-HAP Nanoparticles

The XRD pattern of HAP (Figure 1(a)) revealed distinct peaks at 25.7, 31.9, 32.7, 39.6, 46.7, 49.4, 53.1, and 64.3° owing to (0 0 2), (2 1 1), (1 1 2), (3 1 0), (2 2 2), (2 1 3), (0 0 4), and (3 0 4) planes of HAP, respectively (JCPDS File No. 09-0432) [53]. The CNTs-HAP (Figure 1(b)) exhibited obvious characteristic peaks of HAP. Furthermore, it displayed an additional peak at 42.6° due to the (1 0 0) reflection of CNTs [53]. However, the (0 0 2) reflection of CNTs overlapped with HAP’s (0 0 2) reflection and appeared together at 25.7°. The GR-HAP exhibited the peaks of HAP and another two peaks located at 25.3 and 44.9°, corresponding to (0 0 2) and (1 0 0) reflections of GR, respectively [53]. The characteristic reflection peaks of HAP found in Figure 1(a) were not radically shifted in Figure 1(a–c) for CNTs-HAP and GR-HAP. It elucidates that the phase structure of HAP was not altered in CNTs-HAP and GR-HAP. The SEM image of CNTs-COOH shown in Figure 2a elucidates the effective entangling of CNTs due to –COOH groups. The image of CNTs-HAP (Figure 2b) illustrates HAP’s deposition over CNTs. The deposition of HAP over the entire surfaces of CNTs helped to overcome entanglement and caused efficient exfoliation. Figure 2c shows the loosening of graphene oxide nanosheets and their porous structure due to the opening of planer carbon networks wedged at the edge surface of crystallite by oxidation and exfoliation. These nanosheets have successfully exfoliated into thin nanosheets with wrinkled surfaces. The image of GR-HAP (Figure 2d) indicates the existence of HAP over both surfaces of GR nanosheets. The stacking of HAP-deposited GR nanosheets caused a sandwich-like layered structure. The separation of single-layered, individual GR nanosheets and their exfoliation is visible in Figure 2d. The TEM images of CNTs-HAP (Figure 3a,b) demonstrate the grafting of HAP over CNTs and their distinct visibility. The refluxing of pristine CNTs in the acid mixture caused uneven surface CNTs due to defective sites, which helped in the strong binding of HAP to CNTs. The efficient entanglement of CNTs owing to HAP could be perceptible in Figure 3b. The images of GR-HAP (Figure 3c,d) show HAP’s anchoring on the entire surfaces of GR nanosheets. The wrinkled or crumpled silk wave-like single-layered GR nanosheets are observable in Figure 3c. Figure 3d divulges GR and HAP, and the discrete boundary exists between them. The sequential washings performed with DI, while purifying CNTs-HAP and GR-HAP, could not detach HAP from the surface of CNTs and GR. It shows the resilient adherence of HAP to the surfaces of CNTs and GR, and the strong interaction persists between them. The in situ deposition of HAP over CNTs and GR used in the reported method facilitated sturdy adherence of HAP to the surface of CNTs and GR. In addition, the following preparation method helped in the homogenous distribution of HAP over the entire surface s of CNTs and GR.

### 3.2. Anti-Cancer Activity Screening of CNTs-HAP and GR-HAP

#### 3.2.1. Cytotoxicity Analysis

To verify the potential use of CNTs-HAP and GR-HAP in cancer therapy, their in vitro cytotoxicity was estimated at different concentrations using SUM-159 and MCF-7 cell lines. It has been revealed that HAP has excellent biocompatibility [62]. However, to find the biocompatibility of HAP after conjugating with CNTs and GR, the cytotoxicity of CNTs-HAP and GR-HAP was estimated using an MTT assay. The cytotoxicity was evaluated in the 5–50 μg/mL concentration range. The perceived results for CNTs-HAP are presented in Figure 4. The degree of cytotoxicity of CNT-HAP and GR-HAP was derived after comparing it with that of their respective vehicle controls, i.e., CNT and GR nanotubes. Diminished cell viability demonstrated improved cytotoxicity with increasing incubation time and dose.

CNT-HAP could kill 29.4 and 30.6% of MCF7 and SUM-159 cells, respectively, following a 24 h incubation with a 5 μg/mL dose. A significant time-dependent cytotoxic enhancement was apparent by the treatment of CNT-HAP on the SUM-159 breast cancer cells (41.4%) compared with the MCF7 cells (30.6%). Similarly, the CNT-HAP showed a dose-dependent increase in cell death in both cell lines. Cytotoxicity in MCF7 and SUM-159 cells was recorded as 63.8% and 72.2%, respectively, with 25 μg/mL of CNT-HAP. The 50 μg/mL dose decreased the viability percentage to 11.8 and 10.3% for MCF7 and SUM-159 breast cancer cell lines (Figure 4 and Appendix A).

Following treatment with GR-HAP, a similar dose and time-dependent cytotoxicity trend were observed in each cell. The cytotoxicity of GR-HAP estimated at different conditions is illustrated in Figure 4. At 5 μg/mL of GR-HAP with 24 h of incubation, the viability rate for MCF-7 and SUM-159 cells was 77.0 and 65.9%, with the corresponding death rate of 23 and 34.1%, respectively. The 24 h of treatment data show that GR-HAP is significantly more effective in suppressing the growth of SUM-159 cells than MCF7. The treatment with 25 μg/mL of GR-HAP demonstrated a significant difference between 38.5% cytotoxicity in MCF7 cells and 59.5% cytotoxicity in SUM-159 cells when incubated for 24 h. However, the gap narrowed down, and the cell viabilities were 33.7 and 22.9% in the case of the 50 μg/mL doses for MCF7 and SUM-159, respectively. Cytotoxicity on MCF7 and SUM-159 cells was the highest at 50 μg/mL when treated for 48 h. At 50 μg/mL, it was 8.5 and 13.4% for MCF-7 and SUM-159 cells, respectively.

#### 3.2.2. Clonogenicity Analysis

Clonogenic assays were performed with SUM-159 cell lines for 12 days following 48 h treatment with 5/25/50 μg/mL of CNTs-HAP and GR-HAP to further study the cytotoxic effects over a prolonged period. The SUM-159 and MCF-7 cells treated with CNT-HAP and GR-HAP (at 5 μg/mL, 25 μg/mL, and 50 μg/mL) grew visible colonies after two weeks (image shown for GR-HAP treated SUM-159 cells only). The number of stained colonies revealed that the treatment with both nanoparticle-associated anti-cancer agents decreased the clonogenic ability of the studied cells (Figure 5). These results were consistent with the findings in the cell viability assays. It shows that an increase in the concentration of GR-HAP inhibits the clonogenic ability of breast cancer cells.

#### 3.2.3. Wound-Healing Assay

The wound-healing ability of the cells following CNT-HAP and GR-HAP treatment was evaluated to assess the migration inhibitory potential of the investigated nanomaterials, and the outcomes are summarized in Figure 6. The GR-HAP demonstrated a substantial inhibitory effect on the cell motility of both MCF-7 and SUM-159 cell lines, whereas the CNT-HAP could not inhibit the cell motility comparatively (Figure 6). Figure 6 shows that, in particular, at 25 μg/mL, GR-HAP inhibited cell motility of SUM-159 cells efficiently.

#### 3.2.4. EMT Marker Study

The effect of GR-HAP over epithelial-to-mesenchymal transition (EMT) was determined, and the results are shown in Figure 7. During tumor progression, EMT can be induced by many oncogenic signaling pathways, resulting in epithelial cells losing their cell–cell adhesion and gaining migratory and invasive properties [63]. In consideration of the inhibitory effect of GR-HAP over cellular migration, it was investigated the influence of GR-HAP on EMT marker expression. It revealed that the expression of vimentin (mesenchymal marker), which was high in the GR-treated (Figure 7a) and HAP-treated cells (Figure 7b), was majorly reduced in SUM 159 cells due to the treatment with GR-HAP (Figure 7c). As CNT-HAP did not show any significant inhibitory effect on motility assay; therefore, it was not considered for the immunofluorescence assay with vimentin.

## 4. Discussion and Conclusions

HAP has been studied previously for numerous biomedical applications, such as bone repair [64], and showing great potential in the field of cancer theranostics [25]. Applying different surface modifications allows HAP nanoparticles to be converted into anti-cancer agents, targeting and killing cancer cells without significant adverse effects on healthy cells. Studies showed that HAP nanomaterials inhibit the growth of cancer cells either by targeting mitochondria [31,32] by endocytosis [33], or by generating ROS in the tumor microenvironment [65]. CNTs are also one of the most promising nanomaterials for cancer treatment and therapy [41,42]. Previous studies showed that CNTs could be functionalized with certain functional groups to manipulate the physical and biological properties to kill cancer cells via apoptosis [66]. This study shows the synthesis and characterization of two novel nanomaterials, GR-HAP and CNT-HAP, and their potential anti-cancer properties.

The biocompatible nanomaterials viz., CNTs-HAP and GR-HAP, have been successfully prepared by the grafting of hydroxyapatite (HAP) to carbon nanotubes (CNTs) and graphene (GR) nanosheets. The CNTs-HAP and GR-HAP have possessed substantial in-vitro cytotoxicity against SUM-159 and MCF-7 breast cancer cell lines. The cytotoxicity of CNTs-HAP and GR-HAP was enhanced with increased concentration and incubation time. Compared to CNTs and GR, their cytotoxicity was not significantly enhanced in CNTs-HAP and GR-HAP. It might be because HAP coupled with CNTs and GR has a greater biocompatibility. The cell viability experiment revealed that CNTs-HAP was more effective against SUM-159 cells than it was against MCF-7 cells. A similar trend was observed with GR-HAP, except for 25 and 50 g/mL after a 48 h incubation period. Nevertheless, no systematic link was identified in the cytotoxicity of CNTs-HAP compared to GR-HAP to determine which has a more pronounced effect. These results revealed the significant capability of CNTs-HAP and GR-HAP to alter the cellular viability of SUM 159 and MCF-7 cells to cause severe cytotoxicity.

The high concentration of GR-HAP was more efficient in inhibiting the clonogenic ability of breast cancer cells compared to low concentrations. The GR-HAP demonstrated the inhibitory effect on cell motility of both MCF-7 and SUM 159 cell lines. The expression of vimentin (mesenchymal marker) had reduced in SUM 159 cells by GR-HAP. Overall, this study showed the potential anti-EMT (anti-cancer) properties of GR-HAP in vitro. Further studies must be done to elucidate the studied nanomaterials' anti-cancer role and unearth the molecular mechanism.

## Figures and Tables

**Figure 1 nanomaterials-13-00556-f001:**
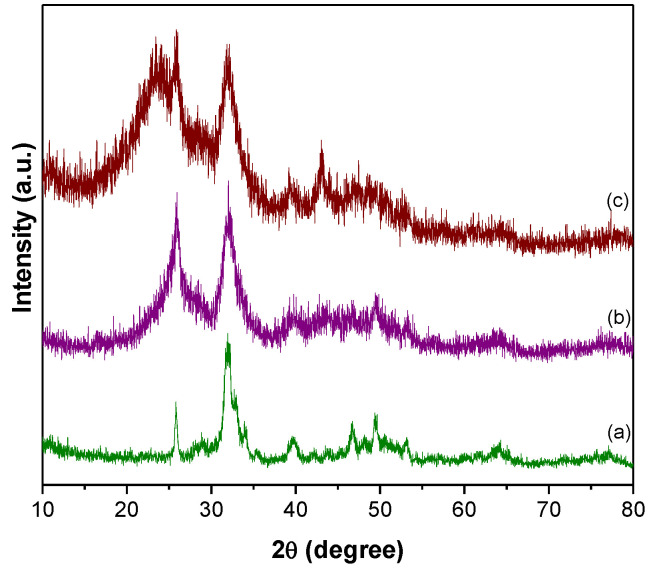
XRD pattern of (a) HAP, (b) CNTs-HAP, and (c) GR-HAP.

**Figure 2 nanomaterials-13-00556-f002:**
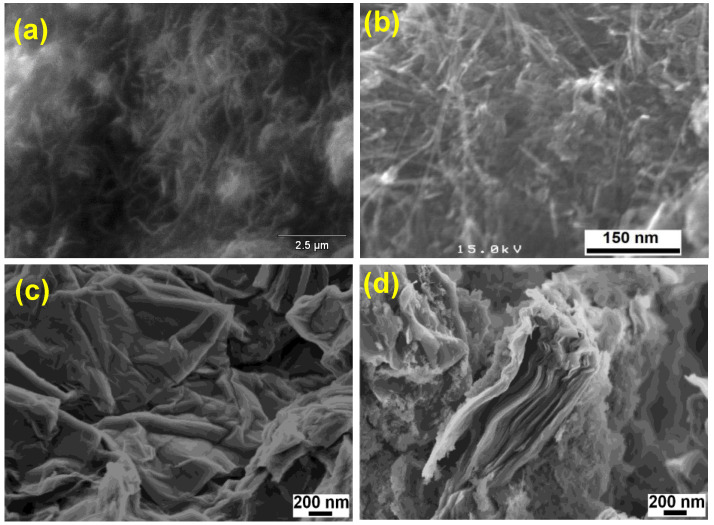
SEM images of (**a**) CNTs-COOH, (**b**) CNTs-HAP, (**c**) graphene oxide, and (**d**) GR-HAP.

**Figure 3 nanomaterials-13-00556-f003:**
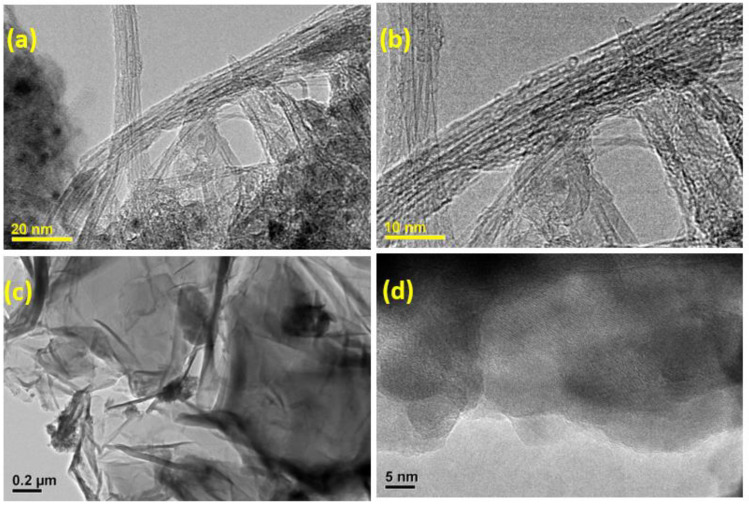
TEM image of (**a**,**b**) CNTs-HAP and (**c**,**d**) GR-HAP.

**Figure 4 nanomaterials-13-00556-f004:**
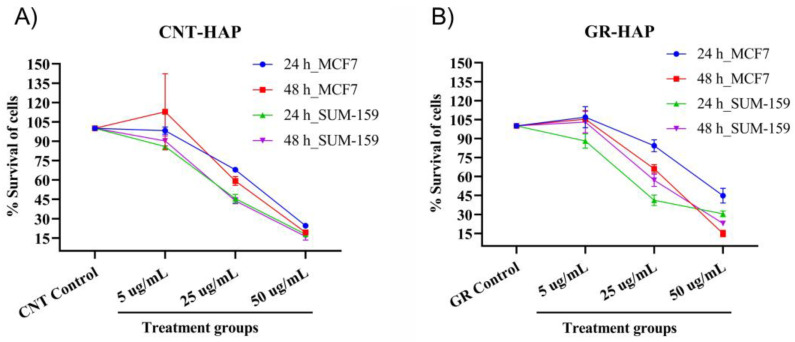
CNT-HAP and GR-HAP suppressed cell survival in a dose and time-dependent manner. (**A**) Cytotoxicity of CNTs-HAP at different concentrations and incubation periods over MCF-7 and SUM-159 cell lines. (**B**) Cytotoxicity of GR-HAP at different concentrations and incubation periods over MCF-7 and SUM-159 cell lines.

**Figure 5 nanomaterials-13-00556-f005:**
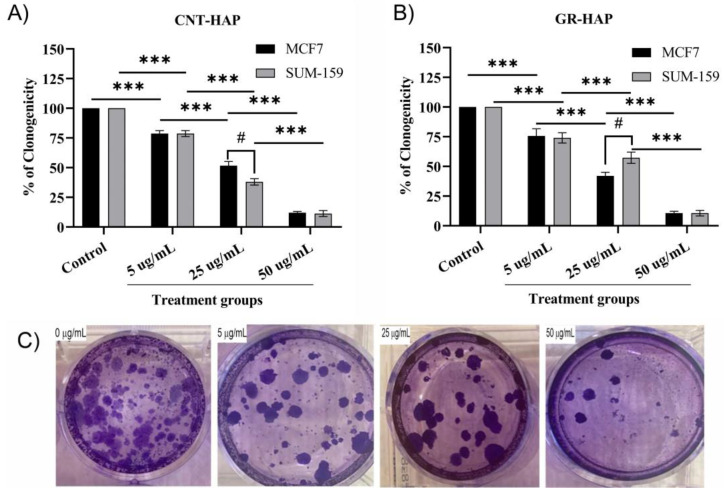
CNT-HAP and GR-HAP dose-dependently inhibited the clonogenicity of MCF7 and SUM-159 breast cancer cells. (**A**) Clonogenic analysis of breast cancer cells after CNTs-HAP treatment. The MCF7 and SUM-159 cells were treated with the indicated concentration of CNTs-HAP, trypsinized, and plated at low density (2000 per well in 6-well plates). (**B**) Clonogenic analysis of breast cancer cells after GR-HAP treatment. MCF7 and SUM-159 cell was treated with the indicated concentration of GR-HAP, trypsinized, and plated at low density (2000 per well in 6-well plates). After two weeks, formed colonies were stained with crystal violet. Clones in a given area were counted for each condition. Columns, mean of three independent determinations; bars, SD. (**C**) After two weeks of treating the SUM-159 cell with 0, 5, 25, and 50 μg/mL of GR-HAP, trypsinized and plated at low density (2000 per well in 6-well plate). Formed colonies were stained with crystal violet. *** *p* < 0.001 indicates a significant change in clonogenicity between different treatment groups, whereas # *p* < 0.05 indicates a significant difference between the two cell lines.

**Figure 6 nanomaterials-13-00556-f006:**
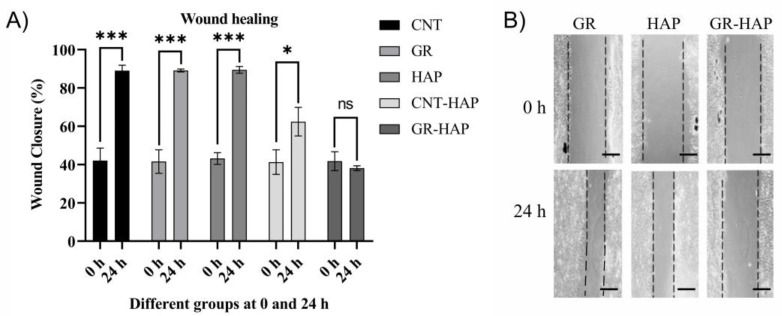
(**A**) Effect of CNTs, GR, HAP, CNTs-HAP, and GR-HAP on wound-healing capacity. Data are presented as means± standard error of the mean of 4-5 independent experiments. (**B**) Representative images of wound healing following exposure to control GR, HAP, and GR-HAP. * *p* < 0.05, and *** *p* < 0.001 represents significant differences obtained from paired *t*-test, whereas ns = not significant. Scale bars measure 250 μM.

**Figure 7 nanomaterials-13-00556-f007:**
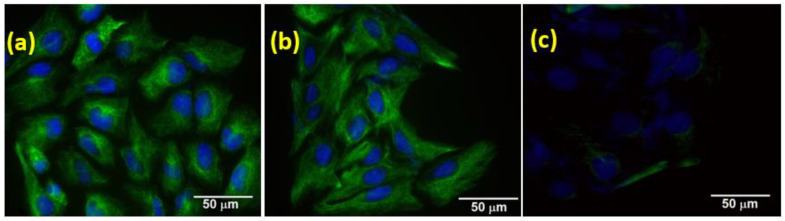
Effect of GR-HAP on EMT. The SUM-159 cells were treated with (**a**) GR, (**b**) HAP, and (**c**) GR-HAP, and vimentin expression was examined using immunofluorescence. Magnification 40X, scale bar: 50 μm.

## Data Availability

Data are available from the authors upon request.

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
