# Peer review of "The Cytotoxicity of Carbon Nanotubes and Hydroxyapatite, and Graphene and Hydroxyapatite Nanocomposites against Breast Cancer Cells"

_nanomaterials, 2023, doi:10.3390/nano13030556_

Round 1
Reviewer 1 Report
In the paper submitted by Nguyen et al., the authors prepared some inorganic systems based on hydroxyapatite, graphene and carbon nanotubes which were further characterized by different techniques in order to demonstrate their ability to kill breast cancer cells.
1. In the introduction section the authors have failed to highlight the originality of the study.
2. No details are provided about the administration route of this system. Which is the administration form? The stability of the systems in an aqueous media was not proven.
3. No information about the cellular viability using human fibroblast cell lines are given. It is well known that both carbon nanotubes and graphene are highly toxic compounds and they are avoided for biomedical applications.
4. No real discussion is given as no comparison with other studies are indicated.
Minor corrections:
5. repetition on the term "leading cause of death" in line 25 and 27
6. tables 1-4 must be moved to Supporting information section as the data is already given in fig 3.
7. line 312: the sentence "compared to pristine CNTs and GR, their toxicity was not significantly enhanced in CNTs-HAP and GR-HAP" is confusing. First of all, the authors have not studied the pristine CNTs and GR. Then, if there are no differences, which is the interest to prepare CNTs-HAP and GR-HAP?
Author Response
Responses to reviewer’s comments
Manuscript ID: nanomaterials- 2133082
Title: The cytotoxicity of hydroxyapatite grafted on carbon nanotubes and graphene against breast cancer cells
We thank the reviewer for their valuable comments and recommendation to accept this manuscript for publication in Nanomaterials. We appreciate the time and effort that the editor and the reviewer have devoted to reviewing our manuscript, and we are grateful for their insightful comments and constructive criticisms. We have taken on board the reviewers’ comments and revised our manuscript accordingly. Below, we have provided a point-by-point response to the reviewers’ comments.
Reviewer-1
In the paper submitted by Nguyen et al., the authors prepared some inorganic systems based on hydroxyapatite, graphene and carbon nanotubes which were further characterized by different techniques in order to demonstrate their ability to kill breast cancer cells.
- In the introduction section the authors have failed to highlight the originality of the study.
Response for 1: The originality of the study is highlighted in the introduction.
- No details are provided about the administration route of this system. Which is the administration form? The stability of the systems in an aqueous media was not proven.
Response for 2: The powdered CNTs-HAP and GR-HAP were dispersed in aqueous media to reach the required concentration, and obtained dispersion is used in experiments. All the aqueous dispersions were stable for the entire experimental period.
- No information about the cellular viability using human fibroblast cell lines are given. It is well known that both carbon nanotubes and graphene are highly toxic compounds and they are avoided for biomedical applications.
Response for 3: The main aim of this study is to investigate the role of the studied nanomaterials in breast cancer cells. Therefore, we have only focused on different breast cancer cell lines and have not used any fibroblasts cell lines for this study.
- No real discussion is given as no comparison with other studies are indicated.
Response for 4: We agree with the reviewers and added a comparison with the previous studies and discussed it.
Minor corrections:
- repetition on the term "leading cause of death" in line 25 and 27
Response for 5: We followed the reviewer’s suggestion and corrected the error.
- tables 1-4 must be moved to Supporting information section as the data is already given in fig 3.
Response for 6: We moved the table 1-4 to the supporting information.
- line 312: the sentence "compared to pristine CNTs and GR, their toxicity was not significantly enhanced in CNTs-HAP and GR-HAP" is confusing. First of all, the authors have not studied the pristine CNTs and GR. Then, if there are no differences, which is the interest in preparing CNTs-HAP and GR-HAP?
Response for 7: We followed the reviewer’s suggestion and corrected the mistake. We removed the term “pristine”. We have changed this discussion part. We have seen cytotoxic effect at a higher concentration. However, GR-HAP showed anti-Epithelial to Mesenchymal Transition properties at a lower concentration. As tumor acquires aggressive and metastatic properties due to EMT, we focused on GR-HAP nanomaterials.

Reviewer 2 Report
In this manuscript, the authors reported the synthesis of CNTs-HAP and GR-HAP hybrids, and further investigated their cytotoxicity against breast cancer cells. The obtained results indicated that both materials exhibited dose-dependent and time-dependent in vitro cytotoxicity towards SUM-159 and MCT-7 breast cancer cells. It is an interesting work. The experiments are good-designed and the obtained results are convincible. In addition, the manuscript is well-organized and good-written. However, there are still some problems that should be addressed carefully. For instance, the material characterizations are weak and the discussion needs improvement. Therefore, major revisions are necessary.
Special comments for the revision:
1. It is necessary for the authors to indicate clearly the novelty and significance of this work by comparing with other released reports with the similar topic. For instance, the authors could discuss the importance of the designed materials and the effects of the materials on cancer therapy.
2. For the design and synthesis of CNTs-HAP and GR-HAP, it is suggested for the authors to add a scheme to indicate clearly the synthesis process of both hybrid materials.
3. The concentrations of CNTs and GR are crucial for the cytotoxicity of the created CNTs-HAP and GR-HAP. However, the effects of different concentrations of materials on their cytotoxicity are not studied by the authors.
4. The morphological characterization of materials should be improved. from Figure 2, it is hard to see the formation of HAP on CNTs and GR nanosheets. The authors are suggested to provide the SEM images of CNTs and GR nanosheets without HAP binding. In addition, to see the HAP clearly, it is necessary for the authors to use TEM to characterize both materials.
5. The synthesis mechanism of HAP on CNTs and GR nanosheets should be discussed.
6. Scale bars should be added to the images in Figure 5b.
7. The authors should indicate clearly the results in Figure 6a, b, and c, respectively. More discussion is needed.
8. The part of discussion is weak. The authors should add more in-depth discussion on the obtained results.
Author Response
Responses to reviewer’s comments
Manuscript ID: nanomaterials- 2133082
Title: The cytotoxicity of hydroxyapatite grafted on carbon nanotubes and graphene against breast cancer cells
We thank the reviewer for their valuable comments and recommendation to accept this manuscript for publication in Nanomaterials. We appreciate the time and effort that the editor and the reviewer have devoted to reviewing our manuscript, and we are grateful for their insightful comments and constructive criticisms. We have taken on board the reviewers’ comments and revised our manuscript accordingly. Below, we have provided a point-by-point response to the reviewers’ comments.
Reviewer-2
In this manuscript, the authors reported the synthesis of CNTs-HAP and GR-HAP hybrids, and further investigated their cytotoxicity against breast cancer cells. The obtained results indicated that both materials exhibited dose-dependent and time-dependent in vitro cytotoxicity towards SUM-159 and MCT-7 breast cancer cells. It is an interesting work. The experiments are good-designed and the obtained results are convincible. In addition, the manuscript is well-organized and good-written. However, there are still some problems that should be addressed carefully. For instance, the material characterizations are weak and the discussion needs improvement. Therefore, major revisions are necessary.
Special comments for the revision:
- It is necessary for the authors to indicate clearly the novelty and significance of this work by comparing with other released reports with the similar topic. For instance, the authors could discuss the importance of the designed materials and the effects of the materials on cancer therapy.
Response for 1: Although several studies have been done using CNT and graphene oxide nanomaterials, the anti-cancer properties of GR-HAP and CNT-HAP have not been studied before. Their role on epithelial to mesenchymal transition (EMT) has not been studied. Therefore, we have focused on the role of those particles in EMT properties in breast cancer cells.
- For the design and synthesis of CNTs-HAP and GR-HAP, it is suggested for the authors to add a scheme to indicate clearly the synthesis process of both hybrid materials.
Response for 2: The scheme depicting CNTs-HAP and GR-HAP synthesis is added to supplementary information and labeled as Scheme 1.
- 3. The concentrations of CNTs and GR are crucial for the cytotoxicity of the created CNTs-HAP and GR-HAP. However, the effects of different concentrations of materials on their cytotoxicity are not studied by the authors.
Response for 3: We indeed studied the effect of CNT and GR on cytotoxicity. At 50 g/mL, the cytotoxic effects of CNT and GR nanoparticles are significantly lower than CNT-HAP and GR-HAP as shown in fig 3 and the figure shown below.
Comparative data showing cytotoxic efficacy of CNT-HAP and GR-HAP. Panel A shows the comparative cytotoxicity of CNT-HAP versus normal control cells, while panel B shows the cytotoxicity of GR-HAP. The individual cytotoxic characters of CNT (C) and GR (D) were analyzed in comparison with that of HAP alone and normal control cells.
- The morphological characterization of materials should be improved. from Figure 2, it is hard to see the formation of HAP on CNTs and GR nanosheets. The authors are suggested to provide the SEM images of CNTs and GR nanosheets without HAP binding. In addition, to see the HAP clearly, it is necessary for the authors to use TEM to characterize both materials.
Response for 4: The SEM images have been improved, and the images for CNTs-COOH (Fig. 2a) and graphene oxide nanosheets (Fig. 2c) have been included in the revised manuscript. Moreover, the TEM images of CNTs-HAP and GR-HAP have been added and labeled as Fig. 3.
- The synthesis mechanism of HAP on CNTs and GR nanosheets should be discussed.
Response for 5: The followed in-situ preparation route used to get CNTs-HAP and GR-HAP, and its mechanism has been discussed.
- Scale bars should be added to the images in Figure 5b.
Response for 6: We have added the scale bar.
- The authors should indicate clearly the results in Figure 6a, b, and c, respectively. More discussion is needed.
Response for 7: We explained the results of Fig 7 (Fig. 6 before revision) and added an extra section in the discussion part.
- The part of discussion is weak. The authors should add more in-depth discussion on the obtained results.
Response for 8: We have added more information in the discussion part.

Reviewer 3 Report
Dear Authors,
Please find my review regarding the paper The cytotoxicity of hydroxyapatite deposited on carbon nanotubes and graphene against breast cancer cells
1. In the results part, subtitle 3.1 says Preparation of the CNTs-HAP and GR-HAP nanoparticles, but the results speak of XRD and SEM. The subtitle for 3.1 must be changed.
2. The SEM images dos not support the finding of authors, that the HAP is covering bot GR and CNT. Further investigation is needed to support this statement. TEM investigation is required to prove that the CNT or GR are decorated with HAP and not just a composite is formed.
3. The rest of the paper is quite good and the conclusions are supported by the data. But at this point the title is misleading because the authors did not prove that the CNT or GR are decorated with HAP. They maybe should change the title and instead of “hydroxyapatite deposited on carbon nanotubes and graphene” maybe as a suggestion “hydroxyapatite carbon nanotubes and hydroxyapatite graphene nanocomposites” can be used
Author Response
Responses to reviewer’s comments
Manuscript ID: nanomaterials- 2133082
Title: The cytotoxicity of hydroxyapatite grafted on carbon nanotubes and graphene against breast cancer cells
We thank the reviewer for their valuable comments and recommendation to accept this manuscript for publication in Nanomaterials. We appreciate the time and effort that the editor and the reviewer have devoted to reviewing our manuscript, and we are grateful for their insightful comments and constructive criticisms. We have taken on board the reviewers’ comments and revised our manuscript accordingly. Below, we have provided a point-by-point response to the reviewers’ comments.
Reviewer-3
- In the results part, subtitle 3.1 says Preparation of the CNTs-HAP and GR-HAP nanoparticles, but the results speak of XRD and SEM. The subtitle for 3.1 must be changed.
Response for 1: The subtitle is changed.
- The SEM images dos not support the finding of authors, that the HAP is covering bot GR and CNT. Further investigation is needed to support this statement. TEM investigation is required to prove that the CNT or GR are decorated with HAP and not just a composite is formed.
Response for 2: To support the evidence of deposition of HAP over CNTs and GR, the SEM images of CNTs-COOH (Fig. 2a) and graphene oxide (Fig. 2c) have been added. In addition, the TEM images of CNTs-HAP and GR-HAP (Fig. 3) have also been included.
- The rest of the paper is quite good and the conclusions are supported by the data. But at this point the title is misleading because the authors did not prove that the CNT or GR are decorated with HAP. They maybe should change the title and instead of “hydroxyapatite deposited on carbon nanotubes and graphene” maybe as a suggestion “hydroxyapatite carbon nanotubes and hydroxyapatite graphene nanocomposites” can be used
Response for 3: The title has been changed to The cytotoxicity of hydroxyapatite grafted on carbon nanotubes and graphene against breast cancer cells.

Round 2
Reviewer 1 Report
The authors have made some modifications but the provided answers are not satisfactory. Even if the main aim was the evaluation of the efficiency against breast cancer cells, it is very important to have some information about the action of the studied system on normal cells; the cancer cells are not living by themselves in the body, they are surrounded by healthy cell lines and if the studied systems kills these cells, then the patient will die faster.
How the stability of the aqueous suspensions was assessed?
Which is the potential administration route?
Author Response

(The authors gave the same response as above.)

Reviewer 2 Report
In this revised version, the authors made suitable modifications according to the comments and suggestions of the referees. I am satisfied with these changes and therefore recommend the publication of this manuscript in current form.
Author Response

(The authors gave the same response as above.)

Reviewer 3 Report
Dear Authors,
All recommendations have been taken into account. So im my opinion the paper can be published as is.
Author Response

(The authors gave the same response as above.)
